# Neurological Mechanisms of Diagnosis and Therapy in School Children with ADHD in Poland

**DOI:** 10.3390/ijerph19137615

**Published:** 2022-06-22

**Authors:** Małgorzata Nermend, Kinga Flaga-Gieruszyńska, Zdzisław Kroplewski, Kesra Nermend

**Affiliations:** 1Department of Early Education, University of Szczecin, 71-004 Szczecin, Poland; malgorzata.nermend@usz.edu.pl; 2Research Team on Civil Procedural Law and Informatization of the Justice, Faculty of Law and Administration, University of Szczecin, 71-101 Szczecin, Poland; kinga.flaga-gieruszynska@usz.edu.pl; 3Institute of Psychology, University of Szczecin, Krakowska 69, 71-017 Szczecin, Poland; zdzislaw.kroplewski@usz.edu.pl; 4Department of Decision Support Methods and Cognitive Neuroscience, Institute of Management, University of Szczecin, 71-004 Szczecin, Poland

**Keywords:** neurofeedback, ADHD, neuroscience, EEG

## Abstract

The paper aims to present a holistic view of attention deficit hyperactivity disorder (ADHD) in pedagogical, psychological, legal, and social dimensions in Polish schools. The authors present the benefits of neurofeedback therapy for elementary school pupils. In order to verify the validity, the paper compares the concordance of a medical diagnosis confirming ADHD syndrome with the occurrence of abnormal electrical brain function recording and abnormalities therein as well as the effectiveness of the neurofeedback therapy. The study confirms that the reported problems faced by pupils and affecting their emotional functioning are reflected in their EEG records. Conclusions from the study lead to the proposal that the neurofeedback assessment should be performed at schools, which should result in the implementation of effective therapy. Moreover, the neurofeedback method should be promoted in Polish schools as an alternative to pharmacological therapy, which, as the research proves, is not always effective. Neurofeedback therapy, similarly to behavioral therapy, is very much needed and useful because it provides optimal conditions for the child’s development and shapes their relations with the environment effectively and harmlessly.

## 1. Introduction

In the era of modern civilization, many factors can be observed that hinder the process of education in schools. Among them are mental disorders in children such as childhood autism, attention deficit hyperactivity disorder, or depression. These disorders have various causes, including genetic, environmental, civilizational, and resulting from individual characteristics [1]. The research conducted under the project “Implementing Mental Health Promotion Action” by Eva Jané-Llopis and Peter Anderson shows that psychiatric diseases contribute to high mortality among children and adolescents more often than other medical conditions [2]. In Polish schools, there is a significant increase in the number of pupils with developmental disorders and difficulties, while attention deficit hyperactivity disorder is one of the most common childhood psychiatric disorders [3,4]. It can be concluded that every teacher has or will soon have a child with such disorders in their class. This is why studies with both theoretical and practical aims are important.

This paper focuses on one of the above problems concerning attention deficit hyperactivity, which is a significant burden not only for the child themselves but also for the school community in which they spend most of their time. Therefore, the research topic undertaken here is extremely important in the context of the child’s proper functioning at school, at home, and in society. Hence, finding appropriate modern practices to counteract this phenomenon is an important issue, as it affects the future life of a young person [4]. In pupils with attention deficit hyperactivity disorder, also called hyperkinetic syndrome [5], the condition can lead to serious problems in functioning at school, which manifest themselves in disobedience, aggression, stubbornness, worse learning results, problems with counting and reading, as well as in relations with peers [3,6]. The functioning of a student with ADHD (attention deficit hyperactivity disorder) in school is associated with inattention, poor planning ability, and impulsivity [5]. This results in the deterioration of the child’s functioning, increasing further with growing external demands, which eventually causes several changes in personal, school, and social performance. A young student, whose personality and identity are being formed, develops erroneous interactions with other people, causing conflicts with the immediate environment, i.e., their parents, teachers, and friends. The consequence of such undesirable behavior can be social marginalization. The students who constantly struggle with failure and lack understanding and acceptance from their immediate environment are at risk of developing depression [7].

Therefore, early intervention in young students is essential to lessen the consequences during adolescence and adulthood. Accordingly, in educational settings, many interventions are made to support the students’ proper functioning. These range from classroom observation, pedagogical diagnosis, and individualized teaching and approach to appropriately targeted measures following the advice of psychological–pedagogical and medical clinics. In the case of learners with this disorder, it is also necessary to introduce such measures that can be implemented within the class team. Individualization of the didactic and educational process, contracts, and recommendations for work with a student should be implemented after the first problems in their functioning have emerged.

The symptoms of ADHD are classified into three groups: attention deficit disorder, hyperactivity, and impulsivity [8]. The diagnostic criteria for these disorders are cataloged in detail in both the DSM-5 and ICD-10 [9]. DSM-5 and ICD-10 are the classifications of diseases, DSM-5 was issued by American Psychiatric Association and relates to mental disorders, and ICD-10 relates to all diseases. There is the newest version of ICD ICD-11, which is not used yet in Poland. According to DSM-5, the diagnostic criteria include inattention, hyperactivity, impulsivity, and symptom appearance not later than in the fifth year of life, while ICD-10 highlights inattention, hyperactivity, and impulsivity. ADHD does not follow the same pattern in children; hence, different subtypes of ADHD are listed (e.g., you cannot ignore the age factor when diagnosing and helping children with ADHD because the course of the disorder changes as the child develops [10,11,12].

ADHD disorders lead to lowering or preventing a child from participating in school activities, which also requires the teachers and school counselors to engage in both a diagnostic process and some forms of therapy. These disorders, being related to the child’s attendance at school, specifically address the following issues. Attention deficit disorders in children are manifested by the inability to focus on a single stimulus, e.g., to concentrate on the teacher’s comment or to complete a task, being easily distracted by various stimuli and situations occurring in the classroom or at the learner’s home. These students tend to postpone or abandon tasks, and they frequently fail to resume them later. They do many unnecessary things in the time allotted for a particular task, they misplace their belongings, and they do not remember what they should do next [13]. Observations made by adults (teachers and parents) reveal that pupils often drift off in their thoughts, especially when they should be focusing on what the teacher is saying or reading. They also have difficulty in reading and counting, which contributes to their learning deficits and poor school performance [12]. They could be involved in a variety of activities such as looking out of the window, playing with school supplies, and tapping objects or their fingers.

Another problem reported by teachers and parents is hyperactivity which is in no way related to the task or activity that the child is currently doing. Their hyperactivity is annoying and tiring to others. Pupils often whirl, run, fidget, and wave their arms or legs without a particular purpose, are unable to sit for a longer time, disturb others, constantly ask questions, do not pay attention to whether it disturbs someone, and often break the rules and regulations established by the educational institution.

Another parallel symptom is hyper impulsivity [14,15]. Acting on impulse, these students do not think about the consequences of their behavior. It is often observed that such children are unable to learn from previous experience and adapt their behavior according to the rules. They rush to answer even though the question is not over; they take risky actions without thinking about whether they are safe. They have problems with waiting for their turn; they feel compelled to do the task here and now. Sometimes they are mean and vulgar to others.

There are many methods of diagnosing and helping children with confirmed ADHD [16]. In the United States, for example, there are many guidelines, both regarding regards diagnosing the disorder and recommendations on how to handle a child in such a situation. These are guidelines for parents, teachers, psychiatrists, as well as for child psychotherapists [17]. Noteworthy are methods that help diagnose ADHD using tools applied in neurology, such as neurofeedback [18,19,20]. These tools can be used both at the stage of diagnosis and therapy. Therefore, teachers and school educators could use them after proper training, which will be proposed in the further part of the article.

Misunderstanding of the behavior of children with problems of this kind often triggers conflicts with other students and their parents. Moreover, teachers and other members of the school staff often behave inappropriately. This, in turn, calls for a more extensive discussion concerning the proper diagnosis of children with ADHD. This, in turn, justifies the necessity to discuss more the proper diagnosis of children with ADHD [21], especially in terms of safeguarding their rights on the grounds of the education law and other regulations protecting individuals with special needs. It refers especially to discriminatory actions because sometimes the problems of children with ADHD are ignored or trivialized at school, and this leads to one of the basic forms of discrimination, which is treating people in different situations in the same way. In particular, this applies to situations when school staff ignores the legal obligation to apply the recommendations contained in the child’s ADHD diagnosis issued by competent authorities, which also includes directives to be respected by the educational institution.

Thus, when pupils are diagnosed with ADHD, many interventions are implemented to support their functioning. One such intervention is pharmacotherapy (used by physicians, esp. psychiatrists), psychotherapy (used by psychologists), and psychoeducation (used in schools). In the school environment, cooperation with parents is undertaken, supporting them in adjusting requirements to the child’s abilities and minimizing the distracting stimuli. It is also important to instruct teachers on how to teach an ADHD child as well as their peers in the classroom [22]. One of the interventions in the case of children with ADHD is pharmacological therapy, which, as research shows, is not always effective. Another measure is behavioral therapy (as a type of psychotherapy), which is extremely supportive of the child’s functioning [23,24]. In recent years, other methods have become very popular and are effective and harmless to the child’s health [25]. These therapies include EEG biofeedback therapy, which aims to optimize brain activity resulting in improved neurocognitive functioning, which in turn leads to improved behavior [3,26].

The paper presents a comprehensive look at the problem of ADHD in children in Polish schools from pedagogical, psychological, and legal aspects. The authors want to show the benefits of using neurofeedback therapy at school. The medical diagnosis of ADHD syndrome was verified with the recording of brain electrical function and disturbances, as well as the effectiveness of using neurobiofeedback therapy was verified [27]. The article focuses on the comparison of reported problems occurring in children and their emotional functioning with EEG records. The research helps to understand how neurofeedback protocols work. Conclusions from this research can be used to make a decision regarding the appropriateness of this research in a Polish school and to conduct effective therapy as a result. The conclusions may also be useful when deciding on the appropriateness of these tests in Polish schools and conducting effective therapy as a result. Since neurofeedback is not known in Polish schools, the authors wanted to present the rationale for its use to help teachers carefully choose the means and methods of work with a child to ensure optimal conditions for their development and formation of relations with the environment.

It should be added here that the importance of wider use of this type of examination in schools is also because, as it has been noted earlier, the ADHD diagnosis in a minor has certain legal implications. This is because a child with ADHD is subject to certain rights and obligations as they require special legal protection from public administration bodies, especially those that execute tasks regarding education and social welfare. In the Polish legal system, two possible legal situations can be distinguished that are relevant for a child diagnosed with ADHD. The first of these is to establish his status as a person with special needs, and the second is to grant to a minor (specifically, a person under 16 years of age as defined in Polish health care law) the status of a person with disabilities. As stated by law, ADHD, often being a co-morbidity, is a premise for granting a disability status. In practice, although the ADHD symptoms alone can become a basis for the child’s disability status only under specific circumstances, in many cases, it is one of the components of a medical condition that is essential to establishing this status.

Polish jurisprudence indicates that an evaluation of the health condition of a person up to the age of 16 to determine whether he or she is a person with a disability and to determine the indications relating to his or her disability (i.e., ways of compensating for it) must take into account the following circumstances [28]. It is first necessary to determine whether such an individual has any of the medical conditions listed in the generally applicable law on criteria for assessing disability in persons under the age of sixteen. Once it has been established whether the individual suffers from one or more of these medical conditions, it is then determined whether they have caused health impairment for more than 12 months and how long they are expected to continue. An assessment is then made as to whether they result in an inability to meet the necessities of life, such as self-care, independent mobility, and communication with the environment, in such a way as to justify the need for total and continuous care or assistance beyond that of a healthy child of that age.

At the same time, the medical opinion mentioned above verifies whether the established conditions entail impairment of bodily functions to such an extent that it requires systematic and frequent therapeutic and rehabilitative interventions at home and outside the home. Moreover, an ADHD diagnosis has important implications for the legal liability of such a person, as exemplified by the definition of the rules of criminal liability. As noted in the jurisprudence, the diagnosis of ADHD results in the fact that the criminal court should raise doubts as to the defendant’s sanity and, as a consequence, declare that the defense of the accused in court proceedings is mandatory [29]. This circumstance is particularly important as regards the proper construction of the juvenile criminal responsibility system. As a consequence, the method in question is of key importance for creating the basis for providing adequate instruments for child protection in many areas of their life. Despite his/her special needs, a child with ADHD has the right to enjoy the social life in its full and to receive educational services, and first of all, to establish proper relations with peers, the education staff, as well as with their family.

## 2. Neurofeedback in ADHD Training

Neurofeedback is a method of modern therapy that is based on the self-regulation of the psychophysical state by using feedback information about changes in selected physiological parameters [30]. Thanks to the feedback, the examined person learns when their brain activity increases in a given frequency band of brain waves and when the bands that have a negative influence on their brain function are dominant [31,32,33]. Although neurofeedback is a modern method of therapy, it is not entirely a new concept. In order to read the frequency bands, an EEG recording is used, which was first made in humans by Hans Berger [34]. Important contributions to the development of research have been made by Joe Kamiya studying Alpha waves [35], Bary Sterman studying SMR (Sensory Motor Rhythm) waves [36], Lubar and Seiferd in the study of Theta and SMR [37]. That research gave rise to many other studies and the wide use of neurofeedback today, including in clinical medicine [38] psychology, and sports [39].

The primary application of EEG is in the medical field, where it is used to detect and analyze EEG images and detect abnormalities, enabling diagnoses to be made by neurologists and technicians [40]. For example, epilepsy can be diagnosed in this way [41,42]. We have entered a new era of neurology where it is possible to view the brain signal directly, in real-time, non-invasively, and without side effects, allowing us to find the true source of problems. Neurofeedback is beneficial in alleviating symptoms of depression [43,44], anxiety disorders [45], developmental trauma and post-traumatic stress disorder [46], autism [47], obsessive compulsions, sleep disorders [48], fatigue, eating disorders, tension, aggression, memory concentration and attention disorders, attention deficit hyperactivity disorder [49,50,51,52,53], dyslexia [31], as well as disorders related to the use of psychoactive substances such as drugs and alcohol [45].

The main purpose of neurofeedback is to teach the subject to self-monitor the brain by measuring brain waves and then providing feedback [31]. Feedback is given with cues that can be audio signals and/or video displayed on a monitor during therapy. After receiving the feedback, the subject can learn to consciously control their brain waves and steer the audio or video displayed. When the band frequency is changed (lowered or raised), the subject’s brainwaves produce various psychophysical symptoms such as calming, lowering anxiety, improving concentration, and reducing tension or stress [54]. These are produced by the activity of brain neurons [55,56]. The image of the signal on the screen that results from brainwave activity takes the form of a line consisting of various waves. When studying the electrical activity of the brain, diverse types and ranges of waves are distinguished: Delta wave with frequency (0.5–4 Hz), Theta (4–8 Hz), Alpha (8–12 Hz), SMR (12–15 Hz), Beta (15–20 Hz) and Beta2 (20–34 Hz). Delta waves are responsible for efficient brain function and regeneration during sleep. Low Delta causes sleep problems, while its high amplitude is seen in people with brain injuries, mental disabilities, and autism. Theta waves are responsible for achieving the flow state. Low Theta causes problems with memory and with achieving the flow state. High Theta causes concentration disorders and loss of attention (so-called “drifting off”).

People whose Theta waves are significantly elevated are characterized by hyperactivity (ADHD). Alpha waves are responsible for, among others, good mood, relaxation, creativity, feeling of calmness, and relaxation or fast learning. Its high amplitude causes attention deficits such as ADD (attention deficit disorder), daydreaming, and excessive fantasizing. Its increased value can also be seen in people smoking marijuana and taking antidepressants. Low Alpha causes psychophysical exhaustion, reduced creativity, learning problems, and inability to relax. SMR waves induce in humans a good resistance to stress, the ability to focus attention for a long time without effort, and in people with epilepsy-good resistance to seizures. Their excessive elevation causes hyper-attention and even anxiety. On the other hand, their low values increase susceptibility to stress, hinder maintaining attention for a long time, and cause psychophysical exhaustion as well as chronic and prolonged stress. The next brain waves that are essential for proper human functioning are Beta waves which are responsible for good attention, efficient cognitive functioning, and easy learning. Elevated Beta waves result in a sensation of over-focus, anxiety, agitation, hyper-attention span, and difficulty concentrating. Conversely, low Beta waves induce learning and attention problems. Beta2 waves provide humans with adequate energy to work and act, inducing peak performance. Decreased Beta2 causes a loss of energy, sluggishness, and slowness, while elevated Beta2 induces over-stimulation and stress. As can be seen from the above description, brain wave frequency disturbances cause fatal difficulties in human functioning. As can be seen from the above description, disturbances in the frequency of waves have adverse effects on human well-being [57]. Different protocols are used for training, which the therapist adapts to the needs of the subject [58].

During neurofeedback treatment, each session is recorded with an electroencephalograph [59]. The result of this continuous monitoring is that the subject is provided with real-time information via feedback so that they can reinforce specific behaviors. The electrical signals from the brain are transmitted in real-time, allowing the subject to learn how to self-control their neural activity through a brain–computer interface [60,61]. When in training, the subject can monitor their progress using a changing image or sound, which can also be rated. Scoring is particularly important in child therapy because it is a mobilizing factor in therapy.

During the sessions, simulation boards, often called games, are used to encourage a child to participate in training. The boards are selected by the child and can be changed when they become bored with a particular game. There are 43 boards available in the EEG DigiTrack^®^ system. The boards contain movable elements that cause changes on the board. The direction of change is either desired or undesired. The description of the steering is adapted to the specificity of the standard protocols, in which one is a promoted rhythm, and the other two are inhibited rhythms except for the Alpha/Theta protocol. The promoted rhythm is responsible for the first steering parameter of the protocol, and the first steered element of the board is associated with it. The inhibited rhythms are associated with the next two steered elements, whereas the Beta2 rhythm is usually the last (third) steering element on the board. It is important to remember to mark whether the parameter is to be promoted or inhibited. The system has two types of boards: continuous and stepped.

Continuous boards are steered continuously. The instantaneous state of the board directly reflects the instantaneous state of the steering parameters. Changes in the board can occur in two directions, desired and undesired, and the changes depend on the magnitude of change in the associated parameter (as in Figure 1). Among the continuous boards, there are also auditory boards where two different sound backgrounds play the key role. In the case of desired changes in the rhythm amplitudes, the sound intensifies while the undesired ones weaken. Each of the backgrounds has a different visualization. The visuals and sounds intermingle. The visuals and sounds depend on the relationship between the rhythm amplitudes and the difficulty crossbars.

The second type of board is step boards, which do not reproduce continuous changes in the steering parameters but increment their state stepwise when the parameters meet certain criteria. Stepping boards in the system depict races. The boards can be steered when the parameters satisfy a criterion (the amplitudes of promoted rhythms exceed the values of the crossbars, the amplitudes of inhibited rhythms fall below the crossbars), and the elements move towards the target (from left to right of the screen), while the unit displacement does not depend on how much the amplitude of the rhythm differs from the value of the crossbar and is always the same. If the parameters do not meet the above criterion, the objects stand still.

Below are two examples of games that children in Polish schools such as to play, along with their brief descriptions. The “clay pot” is a two-parameter game board, which in practice means that two rhythms, the desired and the undesired, are responsible for changing the image. The promoted rhythm is responsible for the amount of fire under the pot (the more, the better), which is the desired change, while the inhibited rhythm is responsible for the amount of steam coming out of the pot (the more, the better), which is also the desired change. If there is little fire and little steam under the pot, the direction of changes is undesirable.

Another game that was often used for therapy was the “Snowman” simulation board.

The action in this board focuses on the snowman’s escape from the evil snowmen; the greater the distance between the escapee and the pursuers, the better the promoted rhythm is responsible for this. The snowman should go as fast as possible—the inhibited rhythm is responsible for this (as in Figure 2). Additionally, on the board, Christmas trees should light up (the more, the better), for which the second inhibited rhythm is responsible. Moreover, the boards produce sounds. These changes are desirable. The undesired changes make the snowman move slowly, “bad” snowmen are close to him, and lights disappear from Christmas trees.

## 3. Materials and Methods

### 3.1. Participants

The pedagogical observation involved 699 elementary school students aged between 7 and 15 years (351 girls and 348 boys) who attended the school at the level of grades 1–8. As a result of pedagogical observation by the pedagogues teaching at the school, students who needed it were qualified for various forms of psychological and pedagogical assistance at school. In case of irregularities, the pedagogues convene a team of specialists and develop a plan of support activities with which parents are acquainted. As a result of teachers’ observations, while conducting classes during the school year, it was noticed that 31 students have symptoms indicating ADHD. Parents and teachers confirmed that it is visible that these students have specific difficulties characterizing psychomotor hyperactivity, i.e., attention deficit disorder, excessive mobility, and excessive impulsivity. These students were qualified for the study. Of the 31 students, the parents of only 9 students chose to be diagnosed by a psychiatrist who confirmed that the students had ADHD. The confirmed diagnosis was then confronted with a neurofeedback EEG study.

The study was conducted on healthy pupils (no bacterial or viral infections were observed) who had not yet been subject to drug treatment related to ADHD therapy. The decision was made by their parents, who were concerned about the side effects of the medication. The pupils joined the study procedure voluntarily and with parental consent. Parents were asked to provide information about the pupil’s ailments, medications being taken, and medical history. The research was conducted in concordance with the rules outlined in the Declaration of Helsinki (2013 version in force) [62]. Before the commencement of the study, the participants and their parents were informed about its course, as well as about the measuring devices used. The pupils were healthy on the day of the study, and their parents gave written consent for their participation in the study and consent for the processing of personal data. The study does not include a classification by the ADHD type as this information was not provided in the medical diagnoses. Parents and pupils were informed before the examination about the measuring devices employed and the course of examination and all their questions and doubts were answered. Detailed interviews with the pupils’ parents were conducted to verify the diagnosis.

Unfortunately, it was not possible to verify the symptoms with the help of neurofeedback because the parents of children were afraid (despite an exhaustive explanation of what the test and training were about) of interference in their children’s brains [6].

The study aimed to compare and verify whether a medical diagnosis confirming ADHD is reflected in abnormalities of neurofeedback EEG recordings. If confirmed, apply therapy and verify its effectiveness. Attention deficit hyperactivity disorder is a very heavy burden on the child. Therefore, an attempt was made to verify whether the use of therapy is justified, as improving the child’s health and functioning is extremely important, even if it was to help only a single child. Trying to help another person is an expression of a humanistic approach to other people and an expression of one’s humanity.

### 3.2. Protocol and Stimuli

The study consisted of two stages. In the first stage, the study was conducted with the eyes open, and in the second stage, with the eyes closed. The closed-eye test is performed because it may show abnormal neurological patterns such as needles. If the results are disturbing, neurofeedback therapy is not undertaken, and the parents of the students are asked to have a neurological consultation and obtain a certificate from the doctor to the effect that neurofeedback therapy is not contraindicated. Open-eye testing, on the other hand, is designed to learn what is happening in the EEG when the student sits quietly and looks straight ahead for several minutes. The participants were asked to look ahead for 3 min and then close their eyes for another 3 min (alpha waves naturally rise with eyes closed). Then they were asked to relax, remain still, and look straight ahead. The electrical activity of the brain was tested using a neurofeedback device from Elmiko to analyze EEG and QEEG records. The QEEG (Quantitative EEG) method is a computerized tool for investigating brain function and dysfunction as well as for planning neurofeedback sessions. The QEEG method permits not only recording the EEG signal but also its quantitative analysis based on a dedicated computer program. Thanks to QEEG, we receive information in the form of numerical results indicating the voltage of a given brain wave and its percentage share in the whole frequency band for particular EEG points.

The examination consisted in placing EEG electrodes on the clean scalp in an international 10–20 system, including frontal lobes, temporal lobes, central plane, occipital lobes, and parietal lobes. Four measuring electrodes were used, which were placed according to the 10–20 standard at locations: Fz, Cz, C3, and C4 and on the right and left ear lobes to determine proper impedance. Proper impedance is one of the most important steps in preparing the patient for electrophysiological signal recording. One of the conditions to obtain a reliable result is a low impedance value in the skin–electrode contact. The impedance values are divided into five ranges. The lowest range (marked in green) is the most desirable; however, the impedance value should not exceed 15–20 k. Throughout the test, it was monitored whether the impedance was appropriate, and its value did not change, as that could indicate that the electrode had detached from the scalp. During the examination, attention was paid to ensure that the subject did not cross their legs, blink their eyes, or look to the side but rather looked straight ahead at a single point and sat still. Following these guidelines helped to minimize artifacts in the EEG recording.

The next step was to manually remove from the recording those artifacts that would interfere with the recording analysis. QEEG assessment was then applied. For the analysis, a 0.3 s high-pass filter, a 40 Hz low-pass filter, and a 50–60 Hz mains filter were set to suppress frequency interference from the power grid. Thus, the analysis results were obtained, allowing the next step of the procedure to be undertaken. The amplitudes of the rhythms and the ratios of the compared rhythms were examined in detail.

The Theta/Beta ratio is a measure that expresses the ratio of slow Theta to fast Beta frequency and reflects the state of active attention [63,64]. If this ratio is high above 3.0, it indicates ADHD syndrome in children and adolescents. It is characteristic of ADHD patients as well as of patients with memory impairment. The second of the ratios, theta/SMR, expresses the ratio of slow-wave frequency to the sensorimotor rhythm. It reflects the state of relaxation with external attention (open eyes) and is particularly disturbed in people with, among others, hyperactivity and emotional disorders and ADHD.

For each participant of the study, after a detailed analysis of the results, a protocol was selected, which determines the desired direction of changes in brain activity. A properly selected protocol allows, during training, to promote one rhythm and suppress another. In the case of a student with ADHD, there is a predominance of Theta, which is the inhibited rhythm, and a smaller Beta which is then promoted rhythm (as in Table 1). Therefore, a Beta/Theta protocol was chosen (promoted rhythm Beta, inhibited rhythm Theta).

The following series of 30 training sessions took place twice a week for 4 months, each session lasting 20 min. All tests were coordinated and conducted by the same therapist. The purpose of the sessions was to provide positive or negative feedback for desired or undesired brain activities.

During the analysis, the focus was on comparing Theta/Beta and Theta/SMR rhythms because, as mentioned before, they are the ones that indicate the presence of ADHD syndrome in a pupil [55].

## 4. Results and Discussion

The study findings confirmed that children diagnosed with ADHD show increased Theta wave activity and decreased Beta or SMR waves in EEG brain activity measurements.

Based on the EEG recordings from points C3 and C4, as well as Fz and Cz, the average amplitude values of each waveband and their percentage shares in the whole spectrum were recorded, along with the average amplitude values for the whole spectrum at each point. These are shown in the study description below, exemplified by five cases.


**Case 1**


In the first case, we present the results of the diagnosis and therapy of a pupil aged 13. The registrations from the central plane point C3-C4 and Fz-Cz (according to the 10–20 system) allow us to conclude that there are abnormal changes in the electrical activity of the brain in relation to the control values. The main one is the increased activity of slow Theta and Alpha waves. Significantly increased Theta/Beta ratios (at Fz = 4.92, Cz = 4.25, C3 = 3.52 and C4 = 3.69) with control values 2–3 and Theta/SMR (at Fz = 4.9, Cz = 4.38, C3 = 3.68 and C4 = 3.74) with control values 1–2 were noted. Elevated theta amplitudes affected all points (Fz = 33.04 μV, Cz = 29.14 μV, C3 = 27.41 μV, and C4 = 24.39 μV), with control values up to 20 μV. The other amplitudes were within normal limits, and no significant asymmetries were seen.

Due to the many abnormalities noted in the electrical function of the brain, a decision was made to lower the amplitude of the Theta waves. The Beta/Theta protocol was used, with Beta as the promoted value and Theta as the inhibited value (settings for Elmiko software).

As a result of the therapy, an improvement in the values at the control points was recorded. The ratios of Theta/Beta (at Fz = 3.07, Cz = 3.3, C3 = 3.11 C4 = 2.78) and Theta/SMR (at Fz = 2.96 Cz = 3.3 C3 = 2.9 and C4 = 2.6) decreased. The Theta amplitude of all points decreased and was at Fz = 23.06 μV, Cz = 25.14 μV, C3 = 24.29 μV, and C4 = 19.8 μV, respectively.


**Case 2**


In another pupil aged 12, recordings from the central plane points C3-C4 and Fz-Cz, also allowed us to conclude that there were abnormal changes in brain electrical activity in relation to control values. The main one is the increased activity of slow Theta waves in relation to fast Beta waves. Elevated Theta/Beta ratios were noted (at Fz = 3.38, Cz = 3.0), with control values at 2–3. Theta/SMR ratio was also significantly elevated (at Fz = 3.65, Cz = 3.47, C3 = 3.27, and C4 = 2.94) with control values at 1–2.

Due to the many recorded abnormalities in brain electrical function, the Beta/Theta protocol was used during training to reduce Theta and enhance Beta.

Significant improvement was noted in the follow-up study. Theta/Beta ratios reached the desired values of Fz = 1.42, Cz = 1.23. There was also a significant decrease in Theta/SMR at Fz = 1.71, Cz = 1.52, C3 = 3.77, and C4 = 1.35.


**Case 3**


In a pupil aged 8, the recordings made from the central plane points C3-C4 and Fz-Cz confirmed abnormalities related to the electrical activity of the brain. The main abnormality was the increased slow Theta activity. Significantly elevated Theta/Beta (at Fz = 4.18, Cz = 4.10, C3 = 3.83 and C4 = 3.68) with control values of 2–3 and Theta/SMR (at Fz = 4.39, Cz = 3.93, C3 = 3.64 and C4 = 3.71) with control values of 1–2 were recorded. Elevated Theta amplitudes were applied to all points Fz = 32.47 μV, Cz = 31.55 μV, C3 = 28.12 μV, and C4 = 30.97 μV, with control values up to 20 μV. No significant asymmetries were observed.

Due to the multiple abnormalities noted in the pupil’s brain electrical function, the Beta/ Theta protocol training was applied. After 30 sessions, improvement was noted, but increased slow Theta activity was still observed. There was a slight decrease in Theta/Beta ratio at Fz = 4.05, Cz = 4.06, C3 = 3.70, and C4 = 3.61 while a decrease in Theta/SMR was recorded at Fz = 3.88. Theta amplitude at Fz decreased by 4.33 μV and was at 28.14 μV, while the difference at C3 was 1.33 μV and at C4- 4.31 μV.


**Case 4**


The next examined pupil was a boy aged 9, in whom the records from the points at the central plane C3-C4 and Fz-Cz also allowed for the conclusion of abnormal changes in the electrical activity of the brain. He showed increased activity of slow Theta waves. Significantly elevated Theta/Beta (for Fz = 4.87 μV, Cz = 4.05 μV, C3 = 3.53 μV and C4 = 3.63 μV) and Theta/SMR (for Fz = 5.01, Cz = 4.00, C3 = 3.61 and C4 = 3.69) were recorded. Elevated Theta amplitudes were applied to all points: Fz = 30.43 μV, Cz= 24.66 μV, C3= 23.47 μV, and C4 = 21.10 μV, with control values up to 20 μV. After treatment with the Beta/ Theta protocol, the values decreased significantly: Theta/Beta for Fz = 3.33, Cz = 3.11, C3 = 2.39, and C4 = 2.81; Theta/SMR for Fz = 3.46 Cz = 2.92, C3 = 2.50, and C4 = 2.76. The Theta amplitude changes applied to all points and were for Fz = 20.03 μV, Cz = 20.87 μV, C3 = 16.43 μV, and C4 = 19.82 μV, respectively, with control values up to 20 μV.


**Case 5**


In the case of the 12-year-old girl, the records from the central sulcus points C3-C4 and Fz-Cz were also abnormal. The Theta/Beta ratio was significantly elevated at Fz = 3.91, Cz = 3.4, with Theta/SMR at Fz = 3.43, Cz = 3.43, C3 = 2.64, and C4 = 2.89. The elevated Theta amplitudes were at Fz = 26.68 μV, Cz = 24.35 μV, with control values up to 20 μV. After treatment with the Beta/Theta protocol, the results improved: Theta/Beta at Fz = 2.96, Cz = 2.94, with the Theta/SMR ratio at Fz = 2.98, Cz = 2.91, C3 = 2.35, and at C4 = 2.55. The Theta amplitudes were at Fz = 16.39 μV and at Cz = 15.88 μV.

What was key to assessing the effectiveness of neurofeedback therapy was the comparison of the indicators concerning the examined pupils. It should be noted that in the article, the selected results were exemplified in order to confirm their relevance for the particular pupil. In all nine cases who were diagnosed with ADHD and underwent neurofeedback therapy, it was noted that as a result of its application, the functioning of these pupils improved. In some pupils, the improvements were satisfactory, while in others less so. That was due to their individual abilities, motivation, and biological conditioning, yet effective neuro-regulation was seen in all students. It should be pointed out that the changes in the children’s functioning were observed by both parents and teachers and were closely related to the lessening of neuropsychiatric disorders in three groups of symptoms: hyper impulsivity, attention deficit disorder, and excessive motor activity.

In regards to the first pupil, whose symptoms significantly affected his behavior and consequently his functioning at school, it was noted that his mental and psychological fatigue decreased. The changes included improved attention, allowing the pupil to focus on the task for a longer period of time, a decreased frequency of forgetting various important instructions and school tasks, an improvement in learning performance, and better grades. In addition, he was less hyperreflexic, which translated into less frequent standing up in class and involuntary disturbances such as tapping a ruler on the desk. His peer relationships also improved as he did not get into conflicts with his classmates as often. Theta/Beta and Theta/SMR ratios were still high but decreased considerably, which bodes well for further therapy.

In the case of the second pupil, it was noted that his capacity increased, thus enabling him to develop his abilities and skills. The pupil was able to concentrate for longer periods, which consequently enabled him to listen effectively to the teacher’s interpretation and to read instructions, tasks, and other texts to the end. The improvement of the pupil’s attention allowed him to avoid mistakes in mathematical calculations (e.g., confusion of signs), which was reflected in his academic performance. In addition, the pupil’s behavior and integration into the class group improved.

The third described pupil turned out to be an interesting case because although the values of amplitudes in points (Cz, Fz, C3, C4) and Theta/Beta and Theta/SMR indices remained high, far exceeding the norms, the effectiveness of the training was evident. In addition, both the boy’s early childhood education teacher and his parents noticed a significant improvement in his functioning in terms of improved attention and reduced hyperactivity, which resulted in better behavior and academic performance. The boy was more relaxed and patient.

The fourth pupil achieved spectacular results because the reduction of Theta waves and indices in his case was outstanding. His therapy brought the desired results, although it should still be continued, of course, due to the elevated Theta/Beta indices at Fz and Cz and Theta/SMR at Fz. The difference between the input and output states was significant. As a result, that translated into his lower aggression towards other children, which in his case posed a severe educational problem, as he was unable to control his emotions and often hit other students. His outbursts of uncontrollable behavior became considerably rarer and ended with verbal conflicts. In addition, his impulsivity decreased. The student showed more willingness to cooperate and take the initiative while previously refusing to follow all the teacher’s instructions. The teachers emphasized that the boy became much less disruptive during lessons, which also benefited other pupils as they could peacefully and consistently follow the curriculum. Even though the student had significant gaps in his education and needed time and effort to make up for them, his academic performance slightly improved. This was due to his improved attention span, which translated into his ability to focus on the lesson for longer.

The last presented case is that of a girl whose parents and teachers reported her severe problems with attention, drifting off in thought and forgetfulness. After the training, the pupil became very active in class, participated in various joint projects, and her grades improved. She could recall what happened during the lesson and was able to talk about it, which had been very difficult before. In addition, she no longer shifted her attention as quickly to other things or situations around her.

Progress was also noted for the other students. As shown, the students’ values decreased but did not improve completely. All students continued with EEG neurofeedback therapy. The goal of further therapy is to achieve the best possible outcome and to maintain self-regulation skills for as long as possible.

In summary, it seems justified to conclude that neurofeedback training in young students with ADHD brings the expected benefits. Improving self-monitoring of brain activity patterns permits slowing down of brain activity in the Theta band and accelerating it in the Beta band. High-frequency waves associated with concentration and neuronal excitability are upregulated [20,21,22]. The research findings presented here support the conclusion that the use of Theta/Beta training in ADHD is valid [63] because it translates into a reduction in the disruptive symptoms of ADHD. For the pupil, this means that they no longer bear the distressing consequences of their attention deficit hyperactivity disorder, which manifests itself in forgetfulness, difficulty in planning and completing tasks, chaotic functioning, unpredictable behavior, hyperactivity, and grabbing the attention of the immediate classroom environment, these consequences being punishment for their “bad” behavior that are the symptoms of the disorder they suffer from. As a consequence, the pupil in therapy gains not only an improvement in their behavior and better academic results, but most importantly, they stop being rejected by their peer group, gain the approval of their parents and teachers, develop self-esteem, and no longer feel alienated. The results were confronted based on the QEEG analysis of the pupils (comparing with norms) and also on the basis of the observations of parents and teachers who noticed a significant improvement in the pupils’ functioning, both in terms of their behavior, attention, and learning outcomes. The student’s functioning in school was discussed in specially convened teams that included the student’s teachers, parents, and therapist.

## 5. Conclusions

This article comprehensively presents the issues related to ADHD both in pedagogical, psychological, and legal aspects present in Polish schools. The obtained results allow concluding that neurofeedback therapy should be conducted in every Polish school because it brings numerous benefits. One of them is the effectiveness of this therapy. What is equally important is that it is accessible to the pupil and can be performed by a therapist, who is usually a teacher who has regular contact with the pupil and their teaching staff. The therapist-teacher has the opportunity to observe the student’s progress not only during the therapy but in the student’s everyday functioning, which is not possible when the therapy is conducted outside the educational institution. Therefore, the teacher can decide whether to modify or continue the therapy depending on the student’s needs. Another significant advantage is that the therapy is free of charge, as it can be provided as part of psychological–educational support at school. The authors have proven that it is possible to confirm the medical diagnosis by EEG testing and effectively apply neurofeedback therapy. Neurofeedback based on standard protocols in attention deficit hyperactivity disorder can be used as a viable treatment alternative. Neurofeedback can be a complementary therapy. However, it should be emphasized that it cannot replace proper treatment because, as research shows, the most effective method of ADHD treatment is still pharmacological therapy [1]. However, it can effectively support it because the program for mapping the electrical activity of the brain can detect the source of abnormalities. The undeniable advantage of neurofeedback is that it is a completely safe method, devoid of side effects.

The school has a significant impact on the educational and emotional development of children and adolescents, but it is also the environment most concerned with mental health problems and the place where they can be diagnosed so that they can be treated early for the benefit of children. Students’ mental health problems can contribute to their poor academic performance, classroom problems, as well as antisocial or health risk behaviors. A holistic school approach to mental health can help reduce the risk of mental health disorders by providing mental health care to support children’s functioning with neurofeedback.

## Figures and Tables

**Figure 1 ijerph-19-07615-f001:**
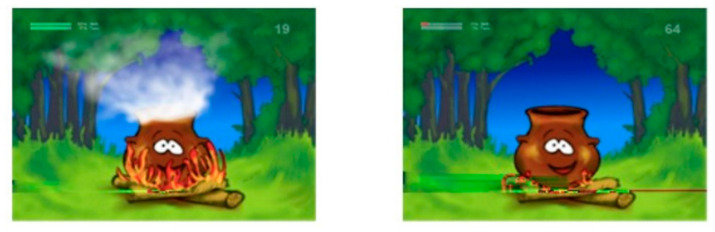
“Clay Pot” simulation board. Source: DigiTrack^®^ EEG System Software.

**Figure 2 ijerph-19-07615-f002:**
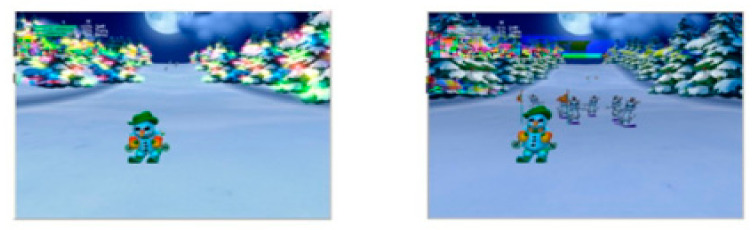
“Snowman” simulation board. Source: DigiTrack^®^ EEG System Software.

**Table 1 ijerph-19-07615-t001:** Range of control values for the ratios of the compared rhythms.

Delta/ Theta	Theta/ Alpha	Alpha/ SMR	SMR/ Beta	Beta/ Beta 2	Theta/ Beta	Theta/ Beta	Theta/ SMR	Alpha/ Theta	SMR/ Beta 2	Alpha/ Beta	Beta/ Alpha
2–6	1.5	1–2	1–1.2	0.5–1	2–3 (children)	1–2 (adults)	1–2	0.5–0.8	0.5–0.8	1–2	0.5–1

## Data Availability

The data relative to the study could be obtained by sending an e-mail to kesra.nermend@usz.edu.pl or malgorzata.nermend@usz.edu.pl Nermend will return directly the files related to the data gathered by the study.

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
