# Peer review of "Neurological Mechanisms of Diagnosis and Therapy in School Children with ADHD in Poland"

_ijerph, 2022, doi:10.3390/ijerph19137615_

Round 1
Reviewer 1 Report
Minor
The protocol manuscript is well written and coherent. The methods and experimental design is well thought, elaborate and well written. The experimental design is in line with the hypothesis. Ther results and conclusions support their conclusion about neurofeedback therapy as a efficient complimentary therapy. I do have some concerns which I listed in point 7. The manuscript needs a little bit more work before it’s ready for publication.
- What is the main question addressed by the research?
The research focusses on the attention deficit hyperactivity which is significant burder for child as well as for the school community in which they spend much of their time.
- Do you consider the topic original or relevant in the field, and if
so, why?
Yes, I find the topic highly relevant in the field. The authors in this work present a comprehensive look at the problem of ADHD in Polish schools and neurofeedback therapy as an efficient complimentary therapy at school.
- What does it add to the subject area compared with other published
material?
Neurofeedback therapy has multiple benefits including it’s effectiveness, can be performed by a therapist at the educational institution and cost effective. . The publications provide evidence supporting neurofeedback therapy as an efficient complimentary therapy in addition to the established effective pharmacological therapy to help individuals suffering from ADHD.
- What specific improvements could the authors consider regarding the
methodology?
The methodology seems coherent with the hypothesis.
- Are the conclusions consistent with the evidence and arguments
presented and do they address the main question posed?
The protocol design is consistent with the hypothesis.
- Are the references appropriate?
Yes. Some are missing reference. Please see section 7 for further information on missing reference.
- Please include any additional comments on the tables and figures.
Please find additional comments and suggestions below:
- Line 72, authors mention DSM-V and ICD-10. In order to reach a broader audience it would be helpful to explain these terms in 1-2 sentences.
- Line 76-77 authors briefly mention that the course of disorder changes as the child develops. It would be list a couple of changes or provide an appropriate reference here.
- Line 100: Hyper impulsivity instead of Hyber impulsivity.
- Line 107 reference is missing. Although the authors provide the references for example of United states, it would be beneficial to provide ”global” reference(s) for line 107.
- Line 129-130: Pleqse explain the terms pharmacotherapy, psychotherapy and psychoeducation or provide a suitable reference for readers to read themselves.
- Line 134: reference missing.
- Line 135: Is behavior therapy same as psychotherapy?
- Line 205: It would be good to introduce SMR (SensoryMotor Rhythm) here.
- Line 212-214: Reference missing.
- Line 228: It would be beneficial for readers if authors could provide a couple of examples of psychophysical symptoms.
- Line 361-362: The explaination of eyes open and eyes closed is missing.
- Line 383:Definition of QEEG (Quantitative EEG) should be introduced when the abbreviation itself is introduced in line #366
- Line 401-402:The criteria for chosing individual protocol is not explained. I am not sure if I somehow missed this information.
- Is it possible to see the results of all 9 selected pupil may be as supplementary data?
- Is it possible to discuss a little bit more about the results and future approached to help Pupil 3 and Pupil 4, both of their indices remain pretty high.
Author Response
We would like to first thank you for your valuable comments and appreciate the time that you spent for reviewing our work. We also admire your vigilance in finding the oversights. We have addressed all your comments as follows and hope you find them satisfactory.
- Line 72, authors mention DSM-V and ICD-10. In order to reach a broader audience it would be helpful to explain these terms in 1-2 sentences.
Response:
Changes to the text have been made (from line 72 to line 82), as suggested.
- Line 76-77 authors briefly mention that the course of disorder changes as the child develops. It would be list a couple of changes or provide an appropriate reference here.
Response:
Inserted reference( line 82)
- Line 100: Hyper impulsivity instead of Hyber impulsivity.
Response:
Text changes made to line 104
- Line 107 reference is missing. Although the authors provide the references for example of United states, it would be beneficial to provide ”global” reference(s) for line 107.
Response:
Inserted reference( line 112)
- Line 129-130: Pleqse explain the terms pharmacotherapy, psychotherapy and psychoeducation or provide a suitable reference for readers to read themselves.
Response:
Changes to the text have been made (from line 133 to line 135), as suggested.
- Line 134: reference missing.
Response:
Inserted reference( line 141)
- Line 135: Is behavior therapy same as psychotherapy?
Response:
Changes to the text have been made to line 140
- Line 205: It would be good to introduce SMR (SensoryMotor Rhythm) here.
Response:
Changes to the text have been made to line 208
- Line 212-214: Reference missing.
Response:
Inserted reference( line 214)
- Line 228: It would be beneficial for readers if authors could provide a couple of examples of psychophysical symptoms.
Response:
Changes to the text have been made (from line 228 to line 229), as suggested.
- Line 361-362: The explaination of eyes open and eyes closed is missing.
Response:
Changes to the text have been made (from line 364 to line 370), as suggested.
- Line 383:Definition of QEEG (Quantitative EEG) should be introduced when the abbreviation itself is introduced in line #366
Response:
Definition was introduced as suggested (line 374-380)
- Line 401-402:The criteria for chosing individual protocol is not explained. I am not sure if I somehow missed this information.
Response:
An explanation have been added (from line 408 to line 413), as suggested.
- Is it possible to see the results of all 9 selected pupil may be as supplementary data?
Response:
The results are included in the table
- Is it possible to discuss a little bit more about the results and future approached to help Pupil 3 and Pupil 4, both of their indices remain pretty high.
Response:
An explanation have been added (from line 551 to line 554), as suggested.
Reviewer 2 Report
The authors embark on a very interesting project of examing various aspects of ADHD as well as methods for improving symptoms via neurofeedback. Although the authors demonstrate great knowledge within the neurofeedback domain, it is difficult to evaluate the scientific soundness of the study as many details are lacking, which ultimately leads to one questioning the conclusions of the study.
Major comments
In general, the introduction lacks a clear structure and hypotheses, is very lengthy, and could easliy be trimmed to avoid repetitions and keep a stringent focus on the subject. There are several very detailed descriptions of, e.g., how inattention, hyperactivity, and impulsivity manifest in the class room, which could be shortened dramatically or even left out.
Likewise, it is clear that the authors have an impressive knowledge about neurofeedback but also this section could be trimmed immensely (shortening or leaving out, e.g., the historical background), while some of the information in section 2 seems more appropriate in a methods section.
The participants section lacks clear in- and exclusion criteria (on what grounds were 31 pupils "qualified" for participation?), a specification of the interview/methods used for verifying the diagnosis, as well as a description of the so-called "pedagogical observation" - what did this observation entail, who performed the observations (what was their training), when was the observations made, etc. Results and points for the discussion should be kept out of the methods section. The authors do not present an analysis plan, and throughout the text, it is unclear which types of analyses have been performed.
The very first sentence of the discussion entails a statement regarding differences in brain waves, when no result supporting this finding has been presented. It is unclear whether this statement is based on a comparison with a control group. If not, it is not possible to make any statements regarding differences - different compared to whom?
The manuscript comments in the discussion on the functioning and behavior of selected pupils, but nowhere do the authors mention how these concepts are assessed. It should be clearly stated what the outcome measures were and how success of the therapy was evaluated.
The entire results section is limited to presenting the results of five pupils and the reader is left to wonder what was found for the remaining participants.
The conclusions do not seem to be supported by the actual findings. The authors state that neurofeedback should be used at every Polish school, which seems bold given the limited data set and unclear presentation of results.
Minor comments
Title: Would change "of School Children" to "in School Children". Also consider removing "in Poland" or simply write "in Polish School Children".
Abstract: The authors refer to psychological, pedagogical, social, and legal perspectives at the beginning of the abstract, but the rest of it focuses on neurofeedback. From the abstract, it is a bit confusing and unclear how and if the remaining concepts are evaluated.
Introduction: Line 51: The abbreviation "ADHD" is used, but has not been introduced previously. Spelled out, it should read "attention-deficit/hyperactivity disorder".
Line 60: Would include a reference for the statement of risk of depression.
Lines 70-78: In my opinion, there two definitions from the DSM-5 and the ICD-10 are very much alike in highlighting inattention, hyperactivity and impulsivity, whereas it appear the authors focus on the differences. I would not call the symptoms "attention deficit disorder", but merely "inattention" to align with "hyperactivity" and "impulsivity". A reference is lacking for ICD-10 and it should be DSM-5 instead of DSM-V. Could the authors please explain what is meant by "holism" as a diagnostic criterion? Finally, the symptom onset has been changed to 12 years in DSM-5.
Line 132: Would never write "ADHD child" - would change to "a child with ADHD".
Materials and methods: Lines 344-350 seem more appropriate in the discussion.
Lines 352-354 describe a result and should not be in the methods section.
Lines 355-359: Unclear what the purpose of this statement is.
Results and discussion: My personal preference is not to mix these to sections and instead have a results section separate from the discussion.
Author Response
We would like to first thank you for your valuable comments and appreciate the time that you spent for reviewing our work. We also admire your vigilance in finding the oversights. We have addressed all your comments as follows and hope you find them satisfactory.
- In general, the introduction lacks a clear structure and hypotheses, is very lengthy, and could easliy be trimmed to avoid repetitions and keep a stringent focus on the subject. There are several very detailed descriptions of, e.g., how inattention, hyperactivity, and impulsivity manifest in the class room, which could be shortened dramatically or even left out.
Response:
This article is intended for a broad audience. The authors realize that the information presented is extensive and detailed, but this was a deliberate intention. It was desired that non-scientists, yes eggs school teachers, could understand the idea of neurofeedback, its origins and how it works. With this basic knowledge, based on scientific facts, the reader will have a basis to trust the method used and implement it in their teaching practice. Often as a therapist I notice that people are afraid of interference in the work of the human brain, therefore it is extremely important to dispel all doubts. It is important to explain comprehensively that this is a safe method used in many different fields including medicine, sports, etc. The authors intend to make the publication available to as wide a range of non-scientist teachers as possible after the article has been accepted for publication.
2.Likewise, it is clear that the authors have an impressive knowledge about neurofeedback but also this section could be trimmed immensely (shortening or leaving out, e.g., the historical background), while some of the information in section 2 seems more appropriate in a methods section.
Response:
The historical background has been shortened changes in lines 206-208, the remaining content according to the authors is consistent with each other and its translation will disrupt the structure of the article.
3. The participants section lacks clear in- and exclusion criteria (on what grounds were 31 pupils "qualified" for participation?), a specification of the interview/methods used for verifying the diagnosis, as well as a description of the so-called "pedagogical observation" - what did this observation entail, who performed the observations (what was their training), when was the observations made, etc. Results and points for the discussion should be kept out of the methods section. The authors do not present an analysis plan, and throughout the text, it is unclear which types of analyses have been performed.
Response:
The description was posted in lines 324-336
4. The very first sentence of the discussion entails a statement regarding differences in brain waves, when no result supporting this finding has been presented. It is unclear whether this statement is based on a comparison with a control group. If not, it is not possible to make any statements regarding differences - different compared to whom?
Response:
The results of the study are included in the review response. Later in the article, the authors present abnormal values in selected students. The statement regarding brainwave differences is justified later in the article. The control group was not included because there are worldwide norms that indicate appropriate values. These values are provided in the article.
5. The manuscript comments in the discussion on the functioning and behavior of selected pupils, but nowhere do the authors mention how these concepts are assessed. It should be clearly stated what the outcome measures were and how success of the therapy was evaluated.
Response:
Explained in line 569-574
6. The entire results section is limited to presenting the results of five pupils and the reader is left to wonder what was found for the remaining participants.
Response:
The article indicates that 9 students were found to have significant impairments. The authors focused on the description of 5, due to the overly extensive description of the studies in the article. The authors wanted to avoid the charge that the description was too extensive, so they decided to select and describe only those cases of students who had the worst outcomes. The remaining students had continued therapy. The authors' next step will be to publish the results of the study after the completion of therapy.
7. The conclusions do not seem to be supported by the actual findings. The authors state that neurofeedback should be used at every Polish school, which seems bold given the limited data set and unclear presentation of results.
Response:
Currently, psychological assistance provided by a psychological therapist is not provided in Polish schools. Psychologists are employed in only a few institutions, usually private ones. Students receive therapy in psychological-educational counselling centres or in other institutions offering such services. The authors came to this conclusion because neurobiofeedback can be used for a variety of psychological-educational problems, including, for example, dyslexia, anxiety, neurosis, etc. Therefore, it would be very widely used at school, and one qualified therapist would be enough to conduct it.
8. Title: Would change "of School Children" to "in School Children". Also consider removing "in Poland" or simply write "in Polish School Children”.
Response:
The title has been corrected.
9. Abstract: The authors refer to psychological, pedagogical, social, and legal perspectives at the beginning of the abstract, but the rest of it focuses on neurofeedback. From the abstract, it is a bit confusing and unclear how and if the remaining concepts are evaluated.
Response:
The authors wanted to present to the reader in a complete way the problem of ADHD, to show what difficulties the students face. It would not be possible without introducing into the article psychological aspects explaining the genesis of ADHD and ways of diagnosing it, as well as legal and social context, as the students are often perceived as dysfunctional children. However, the essence of the article is the diagnosis and therapy of students with ADHD using neurofeedback
10. Introduction: Line 51: The abbreviation "ADHD" is used, but has not been introduced previously. Spelled out, it should read "attention-deficit/hyperactivity disorder".
Response:
Corrected changes are visible in line 51-52
11. Line 60: Would include a reference for the statement of risk of depression.
Response:
Added reference in line 60
12. Lines 70-78: In my opinion, there two definitions from the DSM-5 and the ICD-10 are very much alike in highlighting inattention, hyperactivity and impulsivity, whereas it appear the authors focus on the differences. I would not call the symptoms "attention deficit disorder", but merely "inattention" to align with "hyperactivity" and "impulsivity". A reference is lacking for ICD-10 and it should be DSM-5 instead of DSM-V. Could the authors please explain what is meant by "holism" as a diagnostic criterion? Finally, the symptom onset has been changed to 12 years in DSM-5.
Response:
The authors wanted to introduce the reader to what are the diagnostic criteria, which are very well known among psychologists and psychiatrists.
13. The literature was supplemented line 81. The text was corrected for inattention line 77, DSM-5 was corrected as suggested and ICD-10
Line 132: Would never write "ADHD child" - would change to "a child with ADHD".
Response: is corrected
14. Materials and methods: Lines 344-350 seem more appropriate in the discussion.
Response:
The text is an introduction to the research
15. Lines 352-354 describe a result and should not be in the methods section..
Response:
Description applies to students who qualify for the survey
16. Lines 355-359: Unclear what the purpose of this statement is. .
Response:
The text has been completed 355-357 as suggested
Reviewer 3 Report
Introduction
- Although your introduction is complete, I suggest that you should describe previous experimental studies using Neurofeedback treatment in children with AHDH. It is well known that this treatment has high efficacy, but the readers need to know the context.
Methods
Participants
- I suggest you rewrite how the 9 participants were selected from an extensive sample because it is unclear (indicate which test or surveys were applied, diagnostics, symptoms).
- Provide a table showing the age, scholar grade, and abnormal brain electrical activity of your sample before treatment. Not all subjects with ADHD have electrophysiological abnormalities.
- Eliminate the paragraph located between 344 and 350 lines.
Protocol and stimuli
- The protocol description is incomplete. I suggest you be careful describing all technical details about the treatment:
-Did you do an initial EEG assessing participants’ brain-electrical activity? Did you compare their brain-electrical activity with a normative database? Did you collect changes in all bands throughout the sessions?
-Please check the following paper, they analyze another neurofeedback treatment, but it may help you describe your methods section.
Alatorre-Cruz, G. C., Fernández, T., Castro-Chavira, S. A., González-López, M., Sánchez-Moguel, S. M., & Silva-Pereyra, J. (2022). One-Year Follow-Up of Healthy Older Adults with Electroencephalographic Risk for Neurocognitive Disorder After Neurofeedback Training. Journal of Alzheimer's disease : JAD, 85(4), 1767–1781. https://doi.org/10.3233/JAD-215538
Data analyses
-You did not describe how you analyzed the data. If the analysis was qualitative, you must provide clinical evidence about changes in participants’ symptoms and how these symptoms were associated with electrophysiological data.
Results
-Provide the learning curve for each participant. The reader has to know whether or not the changes in brain electrical activity are associated with treatment or they obey intra-individual electrophysiological variations.
-Provide a table with the brain-electrical activity before and after treatment.
-Provide topographical maps about participants’ electrophysiological changes before and after treatment.
Author Response
We would like to first thank you for your valuable comments and appreciate the time that you spent for reviewing our work. We also admire your vigilance in finding the oversights. We have addressed all your comments as follows and hope you find them satisfactory.
Comments and Suggestions for Authors
Introduction
- Although your introduction is complete, I suggest that you should describe previous experimental studies using Neurofeedback treatment in children with AHDH. It is well known that this treatment has high efficacy, but the readers need to know the context.
Response:
The authors wanted to avoid an overly broad description, so the article cites numerous publications describing neurofeedback therapy in children
Methods
Participants
- I suggest you rewrite how the 9 participants were selected from an extensive sample because it is unclear (indicate which test or surveys were applied, diagnostics, symptoms).
Response:
The text has been supplemented with a description in lines 324-336
- Provide a table showing the age, scholar grade, and abnormal brain electrical activity of your sample before treatment. Not all subjects with ADHD have electrophysiological abnormalities.
Response:
Electrophysiological abnormalities were confirmed for 9 students tested in response attached table
- Eliminate the paragraph located between 344 and 350 lines.
Response:
Paragraph removed as suggested
Protocol and stimuli
- The protocol description is incomplete. I suggest you be careful describing all technical details about the treatment:
- Did you do an initial EEG assessing participants’ brain-electrical activity? Did you compare their brain-electrical activity with a normative database? Did you collect changes in all bands throughout the sessions?
Response:
The description has been supplemented with changes seen in the text, and a table has been attached to the review that includes data for all participants. Data were compared to accepted global norms
Please check the following paper, they analyze another neurofeedback treatment, but it may help you describe your methods section.
Alatorre-Cruz, G. C., Fernández, T., Castro-Chavira, S. A., González-López, M., Sánchez-Moguel, S. M., & Silva-Pereyra, J. (2022). One-Year Follow-Up of Healthy Older Adults with Electroencephalographic Risk for Neurocognitive Disorder After Neurofeedback Training. Journal of Alzheimer's disease : JAD, 85(4), 1767–1781. https://doi.org/10.3233/JAD-215538
Response:
Thank you for your valuable pointing out the article the authors referenced in line 213
Data analyses
-You did not describe how you analyzed the data. If the analysis was qualitative, you must provide clinical evidence about changes in participants’ symptoms and how these symptoms were associated with electrophysiological data.
Response:
A quantitative analysis was conducted based on the QEEG survey (described in the article) and a qualitative analysis where data was obtained during teacher teams (changes seen in lines : 569-574)
Results
-Provide the learning curve for each participant. The reader has to know whether or not the changes in brain electrical activity are associated with treatment or they obey intra-individual electrophysiological variations.
Response:
Because the purpose of the study was to assess the effectiveness of the therapy and not the speed and progress of learning, these data were not archived. It should be noted that each participant's progress in therapy is dictated by the individual's ability to self-regulate learning and each participant's therapy sessions vary. For this reason, follow-ups are performed usually after 30 sessions (this is determined by the therapist). In the following study, 30 sessions were conducted.
-Provide a table with the brain-electrical activity before and after treatment.
Response:
The source data relative to the study could be obtained at: https://doi.org/10.6084/m9.figshare.20015009
-Provide topographical maps about participants’ electrophysiological changes before and after treatment.
Response:
Because the purpose of the study was to evaluate the efficacy of the therapy and not to analyze electrophysiological changes in the form of topographic maps, these data were not archived.
Round 2
Reviewer 2 Report
The authors have addressed the majority of concerns with many of them, however, not resulting in an actual change in the manuscript. Although the focus of the paper is interesting, the paper appears somewhat unstructured and would benefit from a thorough read-through.
The diagnostic criterion "holism" still has not been explained by the authors, and in this reviewer's opinion, it is not acceptable to write, e.g., "ADHD pupil".
Overall, the results hold potential and the paper is a fine contribution to the neurofeedback litterature on ADHD.
Author Response
The diagnostic criterion "holism" was written by mistake during the text edition. Sorry we didint delete it before. Now it is deleted.
- The unfortunate wording of the sentence that was changed in line 51-53.
- The authors also removed lines 357 to 359 as suggested earlier.
- The article is corrected for Englih language by an expert, the latest modifications of the language is shown as blue text throughout the reserach paper.
Reviewer 3 Report
No comments
Author Response
As per the requirement and guidance from the journal, the data should be available online and only the link for online accessibility should be provided within the research article.
Data Availability Statement: The source data relative to the study could be obtained at 617 https://doi.org/10.6084/m9.figshare.20015009